# Channeled ECM-Based Nanofibrous Hydrogel for Engineering Vascularized Cardiac Tissues

**DOI:** 10.3390/nano9050689

**Published:** 2019-05-02

**Authors:** Smadar Arvatz, Lior Wertheim, Sharon Fleischer, Assaf Shapira, Tal Dvir

**Affiliations:** 1School for Molecular Cell Biology and Biotechnology, Tel Aviv University, Tel Aviv 69978, Israel; smadar.klu@gmail.com (S.A.); liorwert@gmail.com (L.W.); fleisharon@gmail.com (S.F.); 0528788@gmail.com (A.S.); 2Department of Materials Science and Engineering, Tel Aviv University, Tel Aviv 69978, Israel; 3The Center for Nanoscience and Nanotechnology, Tel Aviv University, Tel Aviv 69978, Israel; 4Sagol Center for Regenerative Biotechnology, Tel Aviv University, Tel Aviv 69978, Israel

**Keywords:** ECM-based hydrogels, cardiac tissue engineering, vascularization

## Abstract

Hydrogels are widely used materials for cardiac tissue engineering. However, once the cells are encapsulated within hydrogels, mass transfer to the core of the engineered tissue is limited, and cell viability is compromised. Here, we report on the development of a channeled ECM-based nanofibrous hydrogel for engineering vascularized cardiac tissues. An omentum hydrogel was mixed with cardiac cells, patterned to create channels and closed, and then seeded with endothelial cells to form open cellular lumens. A mathematical model was used to evaluate the necessity of the channels for maintaining cell viability and the true potential of the vascularized hydrogel to form a viable cardiac patch was studied.

## 1. Introduction

Hydrogel scaffolds are widely used materials in tissue engineering [1]. This form of material encapsulates the cells, and as opposed to macroporous scaffolds, where the cells are seeded into open pores and may leak out, hydrogel provide real 3D support by 3D cell wrapping [2,3,4,5].

Although this efficient encapsulation may provide the cells with instructive cues for tissue assembly and survival, several challenges still remain. For example, while the cells are encapsulated within a densely packed hydrogel to ensure a proper support, mass transfer into and out of the core of the engineered tissue is limited [6]. Thus, oxygen transport into the tissue may not be sufficient to nourish the inner cells and may result in cell death [7,8]. Furthermore, removal of waste from the tissue is prevented, further harming the cells [9]. To overcome this challenge, prevascularization of the hydrogels is essential [10,11]. Currently, there are two main strategies for engineering blood vessel networks within a parenchymal tissue. The “bottom-up” approach, where the cellular microenvironment induces the formation of a blood vessel network in-between a functioning tissue [12,13,14,15]. Here, endothelial cells and pericytes are co-cultured with parenchymal cells and angiogenic factors. The latter encourage the organization of the endothelial cells into open lumens, stabilized by the pericytes. However, network formation can take a long period of time, risking the survival of the core cells.

In the “top-down” approach, parenchymal-cell seeded hydrogels can be pre-patterned with channels and later on seeded with blood vessel-forming cells. Thus, the core of the hydrogel can be immediately supplied with medium to nourish the entire construct. In recent years, many methods were used in order to create channels within the hydrogel, including soft lithography [16,17], 3D bioprinting [18,19,20], template stamping [21,22,23] and modular assembly [15]. For example, Miller et al. exploited carbohydrate glass to form multi-scale vascular networks using 3D high temperature printing [24]. Choi et al. used a lithographic technique to build microfluidic structures within a calcium alginate hydrogel [16], and Kolesky et al. created a 3D microengineered environments for drug screening using poly(dimethyl siloxane) (PDMS), Pluronic F127 and gelatin methacrylate [19]. However, the challenge of creating complex vascular structures within a thick, biocompatible, homogenous hydrogel still remains.

Another challenge that may arise when using standard synthetic hydrogels or even hydrogels formed from a natural component, is the lack of appropriate biochemical composition that recapitulates the entire natural microenvironment [25,26]. Therefore, extracellular matrix (ECM)-based hydrogels, which are processed by decellularization of different organs were developed [25,27,28,29]. These hydrogels, based on natural ECM, expose the cultivated cells to a wide variety of biochemical content, including fibrous proteins, adhesion molecules and sulfated glycosaminoglycans (GAGs) [30,31]. In addition to the rich repertoire of proteins and sugars that were presented to the cells in the encapsulating microenvironment, the mechanical properties of these hydrogels could be tuned to match those of the natural tissue of interest [28,32]. Thus, Pati et al. demonstrated that encapsulated cells in ECM-based hydrogel can express typical gene markers and assemble into a mature tissue [31]. Dequach and colleagues have shown that the ECM-based hydrogel is able to self-assemble and solidify in vivo, forming a nanofibrous scaffold [25].

Although these engineered materials fairly recapitulate the natural structural and biochemical microenvironment, the use of allogeneic or xenogeneic ECM may hamper the treatment after transplantation due to an immune response to residual antigens (e.g., sugars).

Our group has developed a method to create a patient-specific hydrogel, based on omental ECM. This hydrogel maintains its liquid state at room temperature and physically cross-links in physiological conditions [33]. The omentum-based hydrogel was recently exploited to create patient-specific implants, where both the ECM and the cells can be taken from the patient itself [34]. In this article we report on the development of a channeled omentum-based hydrogel as a proof-of-concept for engineering vascularized cardiac tissues. The omentum hydrogel was mixed with cardiac cells and patterned. Two layers of cell-seeded hydrogels were pasted using the liquid form of the ECM to create a channeled hydrogel, which was later seeded with endothelial cells to form open lumens (Figure 1). Next, a mathematical model was used to evaluate the necessity of the channels for maintaining cell viability. Finally, the potential of the vascularized hydrogel to form a viable cardiac patch was studied.

## 2. Materials and Methods

### 2.1. Omentum Decellularization Process and Hydrogel Preparation

All the incubations during the process were at room temperature (RT) and shaken at 120 RPM unless noted otherwise. Omental porcine tissue was agitated for 1 h in hypotonic buffer (10 mM Tris-HCl, 5 mM ethylenediaminetetraacetic acid (EDTA) and 1 μM phenylmethanesulfonyl-fluoride (PMSF) at pH 8.0). The tissue was then subjected to three cycles of freezing (−80 °C) and thawing (37 °C) using the same buffer. After the last cycle, the tissue was gradually dehydrated by washing it once with 70% ethanol for 30 min and three times in 100% ethanol for 30 min each. Polar lipids of the tissue were then extracted by three 30 min washes of 100% acetone. Subsequently, the a-polar lipids were extracted by three incubations in a 60:40 hexane:acetone solution (8 h each). Then, the defatted tissue was gradually rehydrated and subjected to 0.25% Trypsin–EDTA (solution B, Biological Industries, Beit Haemek, Israel) degradation overnight at RT. The tissue was then thoroughly washed with phosphate buffered saline (PBS), followed by 50 mM Tris buffer with 1 mM MgCl_2_ at pH 8.0. Next, the tissue was gently agitated in a nucleic acid degradation solution of 50 mM Tris, 1 mM MgCl_2_, 0.1% bovine serum albumin (BSA), and 40 U mL^−1^ Benzonase^®^ nuclease (Novagen, Madison, WI, USA) at pH 8.0 for 20 h at 37 °C. Finally, the tissue was washed with a buffer containing 50 mM Tris containing 1% (*v*/*v*) triton-X100 (pH 8.0), subsequently with 50 mM Tris buffer (pH 8.0), three times with PBS and three times with sterile double distilled water (DDW). The decellularized tissue was frozen overnight (−20 °C) and lyophilized. After lyophilization, the decellularized omentum was ground into a coarse powder using a Wiley Mini–Mill and then frozen until further use. Dry, milled omentum decellularized ECM (dECM) was enzymatically digested by adding a 1 mg/mL solution of pepsin (Sigma, St. Louis, MO, USA; 3200–4500 Umg^−1^ protein) in 0.1 M HCl. The final concentration of dECM was 1% (w/v). The dECM was digested for 64–72 h at RT under constant stirring until the liquid was homogenous with no visible particles. Subsequently, the salt concentration was adjusted to physiological levels using PBS×10 and the pH was raised to 7.2–7.4 using 1 M NaOH. In order to solidify the hydrogel, it was incubated at physiological conditions (37 °C and 5% CO_2_ in a humidified incubator) for varying periods of time according to its desired shape and size. The hydrogel was fully characterized previously [33,34,35].

### 2.2. Template Design and Fabrication

The template was designed using Dassault Systèmes’ software—Solidworks. The template was purchased from CALIBER 3D Printing Solution Ltd. (Tel Aviv, Israel). 

The vascular network was designed for maximum native tissue mimicking. Channel geometry design mimics the hierarchical, bifurcating behavior found in biological systems; large channels (parent) bifurcate to form smaller channels (child) that maximize efficient blood flow, nutrient and oxygen transport, and waste removal while minimizing the metabolic cost. In blood vasculature, Murray et al. showed that under laminar flow and conserve volumetric rate, the optimal design equalizes the sum of the radii cubes between the parent and bifurcated child channels [36]. Therefore, we designed the channels in a two-order network architecture that obeyed the Murray law. The vascularized cardiac patch fabrication process started with pouring liquid hydrogel on the template, following by 1 h incubation at physiological conditions for gelation. In order to create closed channels, another solidified hydrogel layer was glued to the channeled layer by liquid hydrogel and the construct was incubated at 37 °C for another hour.

### 2.3. Rheological Properties

The hydrogel rheological properties were examined by Discovery HR-3 hybrid Rheometer (TA Instruments, New Castle, DE, USA) using both Peltier and upper heated plates in order to maintain temperature and 8 mm diameter parallel plate geometry. Prevention of sample evaporation was achieved by DDW drops that were set on the Peltier plate. While examining the gelation process using time sweep, a liquid sample at 4 °C was set in the rheometer. The frequency was set to 1 Hz, the strain to 5% and the temperature to 37 °C. While examining the viscoelastic properties of the hydrogel after the gelation process using frequency sweep, the samples were allowed to gel overnight at 37 °C and then set in the rheometer. The strain was set to 0.63% and the frequency set between 0.01 and 1 Hz.

### 2.4. Histology

Hydrogel samples were placed in OCT (Scigen, Paramount, CA, USA) and frozen in liquid nitrogen. Sections of 60 μm were obtained and affixed to X-tra^®^ adhesive glass slides (Leica Biosystems, Wetzler, Germany). The slides were fixed in cold acetone (−20 °C) for 10 min. Following the fixation, the slides were gradually dehydrated in ethanol (70–100%) and stained with Masson’s trichrome stain (Bio-Optica, Milano, Italy) according to manufacturer’s instruction for collagen detection.

### 2.5. Integration Between the Hydrogel Layers

Two different colors (green and yellow) were added to two different liquid hydrogel samples that were then solidified. Next, a thin layer of liquid hydrogel was poured between the two solidified samples and the construct was incubated for 1 h at 37 °C. The samples were then analyzed using a binocular microscope (NIKON SMZ18) in order to investigate the integration and diffusion between hydrogel layers after adhesion with liquid hydrogel.

### 2.6. Scanning Electron Microscope

Omentum based hydrogel samples were lyophilized and mounted onto aluminum stubs with conductive paint and sputter-coated with an ultrathin (150 Å) layer of gold in a Polaron E 5100 coating apparatus. The samples were viewed under SEM (JEOL Ltd., model JSM-840A, Tokyo, Japan) at an accelerating voltage of 25 kV.

### 2.7. Tensile Testing Instrument

In order to determine the adhesive strength, two glued layers were pulled to the opposite direction by a Llyod mechanical tester, with a 10 N load cell at a rate of 5 mm/min. Every hydrogel layer was glued to a piece of paper that could be gripped by the tensile testing instrument. Next, a few drops of liquid hydrogel were placed on one of the layers, and a different layer was placed on top of it. Following one-hour incubation at 37 °C, the samples were examined. Reference samples weren’t glued and went through the same incubation process.

### 2.8. Mathematical Model

The mathematical model was created by COMSOL Multiphysics 4.3 (Comsol Inc., Burlington, MA, USA). Oxygen concertation profile was calculated based on Michaelis-Menten equation with the following parameters: Maximum cellular O_2_ consumption of 5.44 × 10^−2^ nmol/s/cm^3^, Michaelis-Menten constant for oxygen consumption of 3.79 nmol/cm^3^, diffusion coefficient (oxygen in hydrogel) of 1 × 10^−9^ m^2^/s. Several assumptions were made: (1) no-slip boundary conditions at the walls (2) suppressed backflow (3) all diffusion coefficients are equal and independent of temperature and pressure.

### 2.9. Perfusion

Flow was generated by hydrostatic pressure using a syringe pump (Fusion 200, Chemix, Inc., Staffor, TX, USA) through Fine Bore Polythene tubing (Smiths Medical International Ltd., Kent, UK). The tubing was fitted over the end of a 25-gauge needle on one end and inserted into the inlet on the other end. The flow rate was set to 3 mL/min when dyed liquid was perfused, and 2 mL/day when cell-seeded construct was perfused.

### 2.10. Diffusion Through Channels

To analyze the influence of the channels on the diffusion, hydrogels with and without channels were perfused or placed in a plate with colored fluid. Images were taken at three time points- 0, 90, 180 min. The images were processed in order to determine the color penetration in each of the hydrogel samples using IMAGE J program (NIH).

### 2.11. Cell Culture

Human umbilical vein endothelial cells (HUVEC; passage 4–7; Lonza, Basel, Switzerland) and GFP (green fluorescent protein) HUVEC cells (passage 3–8; Angio-proteomie, Boston, MA, USA) were cultured in endothelial cell medium (EGM-2; Lonza, Basel, Switzerland) at 37  °C in a 5% CO_2_ humidified cell incubator. The cells were suspended in EGM-2 medium in concentration of 2 × 10^6^/mL and seeded inside the channels on channeled hydrogel layer. Next, a thin layer of liquid hydrogel was poured on the channeled layer and covered with solidified cover layer. The whole sample was placed at 37  °C in a 5% CO_2_ humidified cell incubator. After 30 min, the sample was turned over and placed at 37  °C for an additional hour in order to allow the cells to get attached to the top hydrogel layer.

The procedure for cell isolation employed in this study was approved by the Animal Care and Use Committee of Tel Aviv University in Israel (L-11-053). Neonatal ventricle myocytes of 0–3 days old neonatal Sprague–Dawley rats were harvested and cells were isolated by 6 to 7, 30 min cycles of enzyme digestion with collagenase type II (95 U/mL; Worthington, Lakewood, NJ, USA) and pancreatin (0.6 mg/mL; Sigma, St Louis, MO, USA) in Dulbecco’s Modified Eagle Medium (DMEM, CaCl_2_·2H_2_O (1.8 mM), KCl (5.36 mM), MgSO_4_·7H_2_O (0.81 mM), NaCl (0.1 M), NaHCO_3_ (0.44 mM), NaH_2_PO_4_ (0.9 mM)). Following each digestion cycle, cells were centrifuged (600 g, 5 min) and the cell pellet was re-suspended in the culture medium composed of M199 medium (Biological industries, Beit Haemek, Israel) supplemented with 0.6 mM CuSO_4_·5H_2_O, 0.5 mM ZnSO_4_·7H_2_O, 1.5 mM vitamin B12 (Sigma, St Louis, MO, USA), 500 U/mL penicillin and 100 mg/mL streptomycin, and 0.5% (*v*/*v*) fetal bovine serum (FBS). Non-myocyte contaminants were removed by 2 rounds of pre-plating for 50 min on plastic tissue culture flasks in a humidified incubator at 37 °C with 5% CO_2_. Hemocytometer and trypan blue exclusion assay was used in order to examine cell number and viability.

The cells were suspended in 5% FBS M199 medium in concentration of 3 × 10^6^ cells/mL. After that, 1.5% omentum based hydrogel was added in a ratio of 2:1 (hydrogel:medium) and mixed by gentle pipetting. The final concentration of the cells was 1 × 10^6^/mL, and that of the hydrogel was 1% (w/v).

The mixture was poured into a plastic mold with the plastic template in it, and into another plastic mold without a template. The mixture was allowed to solidify for 1 h at 37 °C in a 5% CO_2_ humidified cell incubator. Following incubation, the scaffolds were extracted from the mold and thin liquid hydrogel was poured on the channeled layer and covered with solidified cover layer. The construct was placed at 37 °C in a 5% CO_2_ humidified cell incubator for another 1 h. Next, the sample was placed in a Petri dish with 5% FBS supplemented M199 medium and perfused through the channels at a volumetric rate of 2 mL/day by a syringe pump.

### 2.12. Immunostaining

All the incubations during the process were performed under constant shaking at 60 RPM at RT unless noted otherwise.

Cell constructs were fixed in 4% Paraformaldehyde (PFA) for 30 min without shaking. Following three washes with PBS for 5 min, the cell constructs were permeabilized in blocking solution (1% BSA and 10% FBS) with 0.03% Triton-X100 (Sigma, St Louis, MO, USA) for 1 h and then washed twice in PBS. The samples were then blocked for 2 h in a blocking solution. They were then incubated with primary antibodies to detect α-actinin (1:500; Sigma, St Louis, MO, USA) and CD31 (1:100; Sigma, St Louis, MO, USA) for 2 h, washed three times and incubated for 2 h with Alexa Fluor 647 conjugated goat anti-mouse antibody (1:500; Jackson, West Grove, PA, USA, and Alexa Fluor 488 conjugated goat anti-rabbit antibody (1:500; Jackson, West Grove, PA, USA). For nuclei detection, the cells were incubated for 5 min with 5 µg mL^−1^ Hoechst 33258 (Sigma, St Louis, MO, USA).

Samples were analyzed using inverted fluorescence microscope (Nikon Eclipse, Tokyo, Japan) or confocal microscope LSM 510 Meta (Zeiss, Germany).

### 2.13. Viability Test

The hydrogel samples were crushed and the cells were isolated by enzyme digestion with collagenase type II (95 U/mL; Worthington, Lakewood, NJ, USA) and pancreatin (0.6 mg/mL; Sigma, St Louis, MO, USA) in Dulbecco’s Modified Eagle Medium (DMEM, CaCl_2_·2H_2_O (1.8 mM), KCl (5.36 mM), MgSO_4_·7H_2_O (0.81 mM), NaCl (0.1 M), NaHCO_3_ (0.44 mM), NaH_2_PO_4_ (0.9 mM)) for 1 h in a humidified incubator at 37 °C with 5% CO_2_. Every 15 min, the sample solution was gently pipetted. Following digestion, cells were centrifuged (600 g, 5 min) and cell concentration was calculated using a hemocytometer (Neabauer-improved Assistant Germany) with trypan blue stain for discrimination between viable and dead cells.

### 2.14. Statistical Analysis

Statistical analysis data are presented as means ± s.d. Differences between samples were assessed by student’s *t*-test. *p* < 0.05 was considered significant. ns denotes not significant. Analyses were performed using GraphPad prism version 6.00 for windows (GraphPad Software, San Diego, CA, USA).

## 3. Results and Discussion

To create the hydrogel, omenta of 6 months old pigs were obtained (Figure 2A). Next, the cells were removed and the obtained ECM was lyophilized (Figure 2B) and milled to 1 mm^3^ pieces (Figure 2C). The powder was then enzymatically digested to create a liquid thermoresponsive hydrogel that self-assembles by physical crosslinking at 37 °C (Figure 2D,E, respectively). This behavior is critical to the process of creating the channels within the hydrogel, as the liquid hydrogel will be casted on a patterned template and heated up to solidify, thus forming the negative architecture of the template. The resulted, solidified hydrogel contained nanofibers with an average diameter of 92.18 ± 18.68 nm (Figure 2F,G)

To evaluate the viscoelastic properties of the hydrogel, rheological measurements were carried out (Figure 2H,J). The storage modulus (G’) exhibited a pronounced plateau in the investigated frequency range, with higher values than the loss modulus (Figure 2H), implying that the hydrogel was cross-linked [32]. Moreover, the viscosity of the hydrogel decreased when subjected to a shear strain, suggesting that the hydrogel is shear thinning and can quickly heal after the removal of shear [37]. This property is essential in tissue engineering, as the hydrogel is subjected to shear forces during implantation, and restoration of its original shape is important for proper integration and for tissue regeneration. Furthermore, quick healing of the material would allow the hydrogel to be efficiently patterned after casting, thus accurately forming channels according to the desired 3D topography. Another important property of the hydrogel is its gelation time. As shown in Figure 2J, the hydrogel was terminally solidified after 60 min. This provides the liquid with enough time for casting before solidifying.

We next sought to evaluate the ability of the hydrogel to be patterned and integrated into a 2-layer channeled hydrogel. As a proof-of-concept we chose to pattern simple channels that mimic the hierarchical, bifurcating behavior found in biological systems. These include large channels (parent) that split to form smaller channels (child) [38]. Using a 3D printer, a template with an inlet, followed by two branching points, two convergences and an outlet, was created (Figure 3A). Next, the hydrogel in its liquid form was casted and placed at 37 °C for solidification (Figure 3B). The hydrogel was then removed from the template and the created open channels could be easily observed (Figure 3C). To close the channels within the hydrogel, a few drops of liquid hydrogel, serving as glue, were placed at the edges of the patterned hydrogel, and a second layer of hydrogel without pattern was placed on top. At 37 °C, the 2 layers were glued together to form a channeled hydrogel (Figure 3D).

We next assessed the integration between the two hydrogel layers after using the liquid ECM as a biological glue. Liquid hydrogels were supplemented with either yellow or green color (Figure 3E,F, respectively) and incubated at 37 °C for 1 h for solidification. Next, a thin layer of a liquid hydrogel was poured between the two solidified layers, followed by 1 h incubation at 37 °C. As shown by light microscopy image, the two layers completely integrated with each other without a clear border (Figure 3G). To further support these finding and ensure that the ECM glue did not affect collagen concentration at the border between the layers, the double layered hydrogel was frozen, sliced, stained for Masson’s trichrome and observed under the microscope. The border between the layers could not be seen and no changes in collagen density could be observed (Figure 3H). SEM images of the integration point further indicated on the internal morphology of the ECM-based biological glue (Figure 3I). Finally, to assess the tensile strength which indicates on the physical integration between the two layers, a mechanical tester was used. The two layers were stretched to opposite sides and the minimal force needed for detachment was measured. While the non-glued layers did not adhere to each other, the force needed to dissociate the integrated hydrogel was 52.33 ± 3.712 mN (Figure 3J).

The channels within the hydrogels were created to enhance mass transfer into the core of the hydrogels. Without efficient supply of oxygen and nutrients the cells within the hydrogel will not survive [8]. Therefore, our next step was to predict oxygen gradients within the cell-encapsulating hydrogel by a mathematical model. The variable parameters, such as cell density, fluid flow and channel geometry were identical to those of the experimental studies. Diffusion coefficient factor of oxygen in a hydrogel was taken from Mattei et al. [39], and oxygen consumption rate by the cells was calculated using Michaelis-Menten equation as previously described [40]. As shown in Figure 4, creating the channels within the hydrogel significantly enhanced oxygen transfer into its core. The channeled and pristine hydrogels were divided into 5 z-sections, and oxygen concentration was assessed at the initiation of the flow (*t* = 0) and at steady state (*t* = 90 min). While oxygen could be transferred by convection and diffusion to the periphery and the core of the channeled hydrogel, oxygen diffusion was limited to the outskirts of the pristine hydrogels, leaving an oxygen-free core (Figure 4A,B). Analyzing oxygen transfer at the middle section, before steady state is achieved (t-45 min) revealed an area with a very low oxygen concentration at the pristine hydrogel (Figure 4C). Contrary, in the channeled hydrogel, smaller areas were exposed to low concentrations of oxygen (Figure 4D). This implies that immediately after cell seeding and layer integration, the channels in the gels provide an efficient oxygen transfer to the cultured cardiomyocytes, and may help to maintain cell viability. Calculating the average and minimum concentration of oxygen over time revealed higher levels at the channeled hydrogels that are sufficient to maintain cardiac tissue viability (Figure 4E,F) [41]. As mentioned previously, the location, geometry and volume coverage of the channels within the hydrogel were designed to prove the concept of their necessity. Future designs should be optimized to provide the most efficient delivery of oxygen throughout the hydrogel.

We next sought to evaluate the integrity of the channels and the ability of the fluid to flow within the hydrogel without leaking in-between the layers. A tube was inserted into the hydrogel and dyed medium was flown at a rate of 3 mL/min. As shown, the medium filled the channels and flown toward the outlet without leakage in-between the layers (Figure 5A). This provides further support that the liquid ECM acts as efficient biological glue after heating, integrating the two layers without blocking the channels. To analyze the penetration of solutes into the core of the hydrogel, channeled or pristine hydrogels were exposed to dyed liquid. An inlet tube was inserted into the channeled hydrogel to provide perfusion (Figure 5B). Following, the liquid was added to a petri dish containing the two samples. As shown, after 90 min, the channeled hydrogel was filled with the dye, while in the pristine hydrogel, the dye could only diffuse into the outskirts (Figure 5C). After 180 min, the dye continued to diffuse into the core of the pristine hydrogel, however a large, weakly stained area could still be observed (Figure 5D). The lack of staining in the core of the pristine hydrogel and the quick penetration of the dye by convection and diffusion into the channeled hydrogel emphasize the need for perfusion in order to nourish the inner cells with oxygen and nutrients.

We next sought to evaluate the assembly of the vascularized cardiac patch. Cardiac cells were isolated from neonatal rat ventricles and mixed with the liquid hydrogel. The mixture was either poured onto a patterned cast or onto a flat one, followed by 1 h incubation at 37 °C. The layers were then removed from the template and glued with the liquid ECM as described before. Following, either GFP-expressing or normal endothelial cells were perfused into the channels and were allowed to assemble into a tube. On day 4, the engineered tissue was observed under a fluorescent binocular. As shown in Figure 6A, the GFP-expressing endothelial cells assembled within the channels. Confocal microscopy revealed that the endothelial cells have formed an open lumen, allowing medium perfusion (Figure 6B). Immunostaining of the endothelial cells on the surface of the channels with anti-CD31 antibody (pink) revealed that the cells formed both cell-cell and cell-matrix interactions, forming a homogenous, tightly-packed cell layer (Figure 6C). Such organization on the hydrogel surface will ensure medium flow within the channels without leakage. Future experiments should involve a co-culture of endothelial cells with pericytes [42]. To evaluate cardiomyocytes presence within the vascularized hydrogel, the cell-seeded constructs were immunostained for cardiac sarcomeric actinin (pink), while endothelial cells expressed GFP. As shown, cardiac cells were homogenously dispersed within the hydrogel around the endothelial cell lumens (Figure 6D). Higher magnification of the cardiac cell area revealed elongated cardiac cell bundles with massive actinin striation (Figure 6E).

Finally, we sought to evaluate whether fabricating channels within the hydrogel and perfusing medium contribute to cell viability. Cardiac cells were seeded in pristine or channeled hydrogels. The latter were either perfused or statically cultured. On day 4, cell viability was measured and normalized to day 0. As shown in Figure 6F, cell viability in the hydrogel with the perfused channels was significantly higher than the viability in the pristine hydrogel (*p* = 0.002) and the non-perfused channeled hydrogel (*p* < 0.0001). Interestingly, comparing between the non-perfused channeled to the pristine hydrogel revealed no statistical significance, indicating on the importance of fluid flow through the hydrogel to nourish the cells.

## 4. Conclusions and Future Directions

We have shown the ability to fabricate channeled-ECM-based hydrogels to enhanced mass transfer into cultivated cardiac cells. The hydrogel was composed of a patterned layer and a flat layer, glued together to form channels. Both layers encapsulated cardiac cells. Endothelial cells were seeded within the channels to create open lumens, which allowed transfer of liquids.

Looking forward, future theoretical and experimental studies should focus on the optimization of the channels in the 3D hydrogel to achieve ideal transfer of oxygen. Moreover, more efficient approaches to fabricate the cell-seeded lumens should be explored. One of these approaches may be 3D printing of vascularized cardiac patches from the ECM-based hydrogel. Finally, efficient methods to deliver these vascularized tissues by injection to the infarcted heart should be developed.

## Figures and Tables

**Figure 1 nanomaterials-09-00689-f001:**
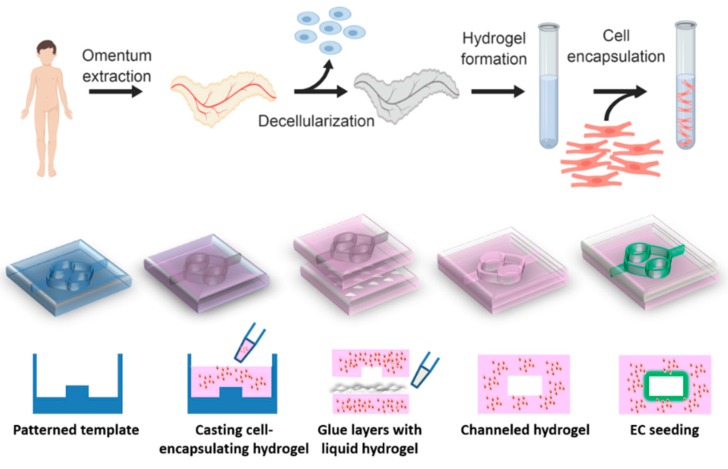
Schematic overview of the concept. Omental tissue is isolated from the patient and undergoes a quick decellularization process to produce omentum-based thermoresponsive hydrogel. The hydrogel is mixed with cardiac cells and then patterned. Two solidified seeded-layers are integrated together using the liquid form of the hydrogel, to create open lumens, which are later seeded with endothelial cells to create a vascularized cardiac tissue.

**Figure 2 nanomaterials-09-00689-f002:**
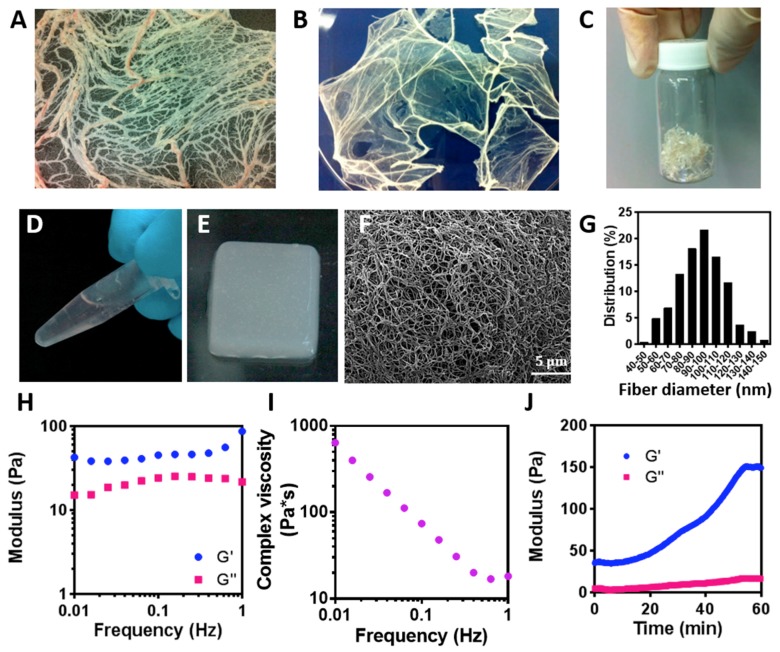
Hydrogel preparation process and characterization. (**A**) Fresh omentum, prior to decellularization process. (**B**) Omentum after complete decellularization and lyophilization. (**C**) Milled decellularized omentum powder. (**D**) Omentum-based hydrogel at room temperature. (**E**) The omentum-based hydrogel under physiological conditions. (**F**) SEM of the nanofibrous structure of the hydrogel. (**G**) Fiber diameter distribution. (**H**–**J**) Omentum-based hydrogel rheology. (**H**) Storage G′ (blue) and loss G″ (pink) modulus versus frequency. (**I**) Complex viscosity of omentum-based hydrogel versus frequency plots. (**J**) Representative curves of storage (G′; blue) and the loss modulus (G″; pink) during gelation at 37 °C.

**Figure 3 nanomaterials-09-00689-f003:**
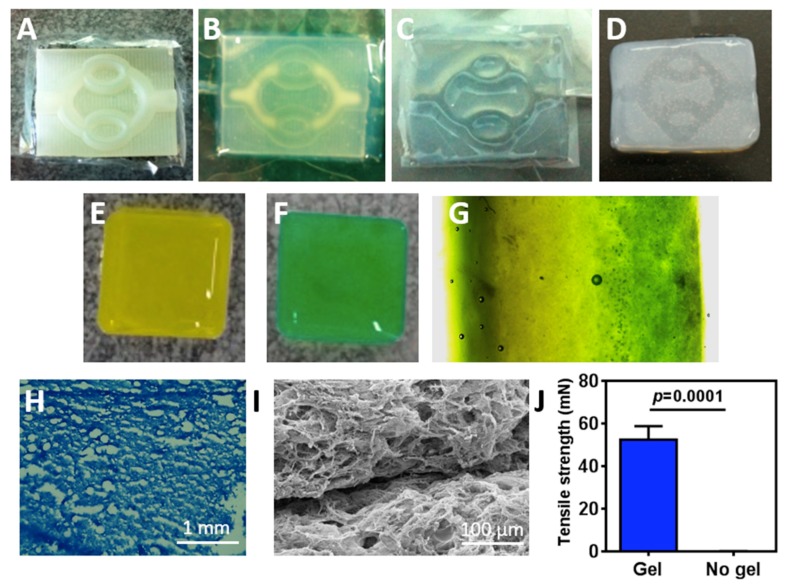
Fabrication and integration of the channeled hydrogel. (**A**) The template. (**B**) The casted hydrogel on the template. (**C**) The channeled hydrogel. (**D**) The entire closed channeled hydrogel after the flat top hydrogel layer was glued to the open channel layer by the liquid ECM solution. Two layers of flat hydrogel colored in different colors: (**E**) Yellow color (**F**) Green color. (**G**) The integration area between the hydrogel layers. (**H**) Masson’s trichrome (Collagen) staining in the integration zone between the hydrogel layers. (**I**) Internal morphology of the integration area between the hydrogel layers as visualized by SEM. (**J**) Tensile strength of two hydrogel layers with and without a glue hydrogel layer.

**Figure 4 nanomaterials-09-00689-f004:**
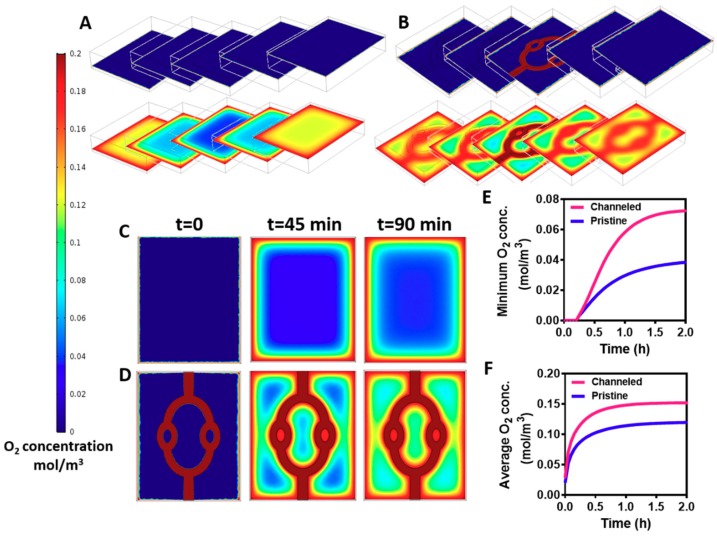
A mathematical model of oxygen transfer within the hydrogel over time. (**A**) Pristine hydrogel. (**B**) Channeled hydrogel. The hydrogels (both channeled and pristine) were divided into five *z*-axis sections. Oxygen concentration at *t* = 0 and *t* = 90 min (steady state) in pristine and channeled hydrogels was modelled. Oxygen concentration as modelled in the core of the hydrogel. (**C**,**D**) Oxygen concentration at *t* = 0, 45 and 90 min (steady state) in pristine (**C**) and channeled hydrogels (**D**). (**E**) Minimum oxygen concentration in pristine (blue) and channeled hydrogels (pink). (**F**) Average oxygen concentration pristine (blue) and channeled hydrogels (pink).

**Figure 5 nanomaterials-09-00689-f005:**
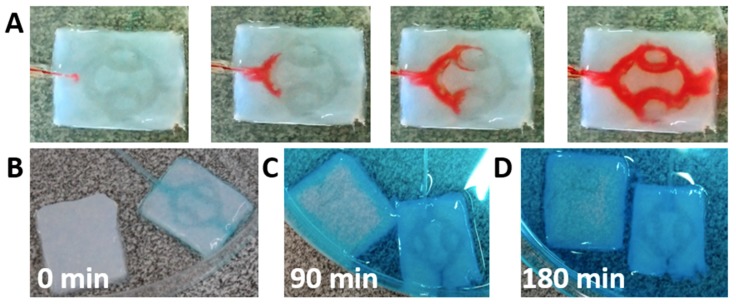
Perfusion and diffusion in the channeled hydrogel. (**A**) Fluid flow through the hydrogel channels. (**B**) Dye diffusion after 0 min, (**C**) after 90 min and (**D**) after 180 min in pristine-static and channeled-perfused samples.

**Figure 6 nanomaterials-09-00689-f006:**
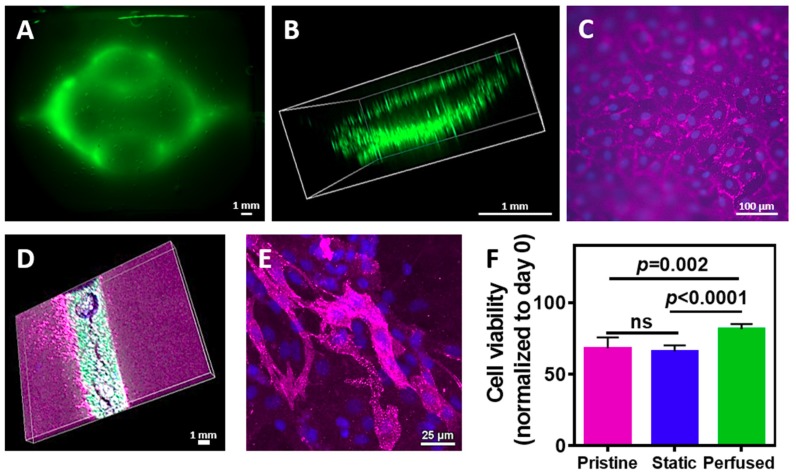
Cells within the patterned hydrogel. (**A**) GFP-expressing endothelial cells were attached to the hydrogel channels. (**B**) Confocal image of a lumen composed from the seeded endothelial cells. (**C**) Endothelial cell layer on the channel wall (pink: CD31, blue: nuclei). (**D**) Co culture of endothelial cells seeded in the channel and cardiac cells in the bulk. (**E**) Cardiac cell assembly in the omentum-based hydrogel (pink: cardiac sarcomeric actinin, blue: nuclei). (**F**) Viability of cardiac cells in pristine hydrogel and vascularized hydrogels with and without perfusion.

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
