# Peer review of "Channeled ECM-Based Nanofibrous Hydrogel for Engineering Vascularized Cardiac Tissues"

_nanomaterials, 2019, doi:10.3390/nano9050689_

Round 1
Reviewer 1 Report
Overall, the manuscript carries an excellent idea. The article has been well thought and written. The data is mainly good and many readers might be interested in these results. The paper is interesting and worth publishing.Author Response
Attached

Reviewer 2 Report
The authors reported the development of a channeled ECM-based nanofibrous hydrogel for engineering vascularized cardiac tissues. The vascularized tissues increased mass transfer to the cells and maintained the cell viability. By mixing cardiac cells with omentum hydrogel in desired pattern, they created channels foe seeding endothelial cells in the constructs. They also proposed a mathematical model to simulate oxygen transfer within the hydrogels, which showed the oxygen gradient within the hydrogel over time. The works is interesting and shows further potential of hydrogel for engineering vascularized cardiac tissues patch. Here are my comments on the work.
-The abstract is short. More quantitative results should be added to the abstract.
-Some details of simulations, such as boundary condition and number of computational grid should be mentioned.
-Figure 3 is a bit confusing. Please add more text to the Figure.
-The quality of Figure 6C is not good.
-It would be good to cite more recent works on cardiac tissue regeneration in the Introduction.
Reviewer 3 Report
Comments to the authors
In this paper, the authors describe the process of preparing a three-dimensional scaffold consisting of a hydrogel on which cells were seeded. The paper might be interesting but many issues are present. To name two: i) the characterization of the proposed biomaterial is very lacking and ii) the route of administration and the purpose for which it was designed are not clearly indicated. Therefore, I suggest to reject this paper. Please find below my specific comments.
- Abstract. It is to be completely rewritten because it is not representative of the main text. Moreover, what is an "omentum hydrogel"? Omentum is a layer of peritoneum: and the hydrogel is a colloidal system formed by polymeric chains dispersed in water. What does an anatomical district have to do with a colloid? Again, it is not clear from the abstract how the construct is prepared and the results and conclusions are missing.
- Introduction section. You stated “Our group has developed a method to create a patient-specific hydrogel” what does it mean? Does it mean that I have to do an operation and take a piece of Omentum? And would the cells instead be allogeneic? Moreover, the route of administration of the construct is not clear. In the conclusion section, you write that it would be injectable but with which needle? What size is the three-dimensional construct?
- Figure 1. Each step needs to be better explained. Letters should be included to explain each step.
- Materials and methods. The experimental design is missing: you have to write in detail what has been done and why.
- Materials and methods. The characterization of the hydrogel is completely missing. You only report rheology study and SEM which are not enough. A physico-chemical characterization, the permeability to model molecules (nutrients?), the biodegradation test, the residual solvents determination and biocompatibility/immunogenicity tests are completely missing. Moreover: was the Omentum of human or animal origin? In Figure 1 a human body is reported, should I deduce that they are human? If so, have been all the procedures approved by an ethics committee?
- Materials and methods, cell culture paragraph. Here it turns out that you worked with rat cells. Again I don’t know if the Omentum was human or murine.
- Results and discussion. It seems to me that this is a physical hydrogel even if it is not specified: what is the biodegradation time of the scaffold in an aqueous medium? Is it sufficient to have an implantable pseudo-tissue formed in vitro?
- Results and discussion. Here it turns out that the Omentum is of a pig. You have to write it also in the material and method section.
- Results and discussion. Has the qualitative-quantitative composition of the hydrogel been investigated? What is it made of? Have the residual organic solvents been determined? There is no physico-chemical characterization of the product, only viscosimetry. Is the production reproducible batch to batch? How many tests have been performed?
- Results and discussion. Only here we understand that this work is a proof-of-concept. You must highlight it at the beginning.
- Figure 2. What do you mean by physiological conditions? Temperature? pH? Ionic strength? The chemical composition of the medium? In figure G I do not read what is on the axis.
- Figure 3. The dyes you used diffuse because they are soluble in water so it is obvious that you cannot see a clear separation between the two colours. In my opinion, this method does not prove that the two materials have been integrated but only that the dyes diffuse. If the authors do not find solid justification for this objection they must remove the figure of the colourations. In fact, from the SEM it is very clear that the two layers are not integrated.
- Results and discussion. You state “Future experiments should involve a co-culture of the ECs with pericytes [38]”. Have future experiments already been published in ref 38?
- Conclusions. Of 3D printing? It appears here for the first time.
Round 2
Reviewer 3 Report
no comments
Author Response
/